# Emotion Regulation in Toddlerhood: Regulatory Strategies in Anger and Fear Eliciting Contexts at 24 and 30 Months

**DOI:** 10.3390/children10050878

**Published:** 2023-05-14

**Authors:** Silvia Ponzetti, Maria Spinelli, Gabrielle Coppola, Francesca Lionetti, Giulio D’Urso, Prachi Shah, Mirco Fasolo, Tiziana Aureli

**Affiliations:** 1Department of Neurosciences, Imaging and Clinical Sciences, “G. D’Annunzio” University of Chieti and Pescara, Via dei Vestini 33, 66100 Chieti, Italy; s.ponzetti@unich.it (S.P.); francesca.lionetti@unich.it (F.L.); giulio.durso@unich.it (G.D.); mirco.fasolo@unich.it (M.F.); tiziana.aureli@unich.it (T.A.); 2Department of Education, Psychology, Communication, “Aldo Moro” University of Bari, 70121 Bari, Italy; gabrielle.coppola@uniba.it; 3Department of Pediatrics, University of Michigan Medical School, Ann Arbor, MI 48109, USA; prachis@med.umich.edu

**Keywords:** emotion regulation, toddlerhood, anger, fear, longitudinal study

## Abstract

The study investigated the emergence of toddlers’ regulatory strategies in aversive contexts. Forty-two toddlers were observed at 24 and 30 months of age using two paradigms designed to elicit fear and anger. We examined toddlers’ use of regulatory strategies at these two stages of life regarding the frequency of self-versus other-oriented strategies and of reactive versus more controlled behaviors. Results showed that the type and level of control of strategies used in toddlerhood in managing negative emotions depend on emotion (e.g., fear versus anger) and age. Toddlers used self-oriented strategies to regulate fear and other-oriented strategies to regulate anger. To manage fear, when toddlers got older, they increased the use of reactive strategies (i.e., releasing tension) and decreased the use of more purposeful strategies (i.e., dealing with the aversive stimulus). In contrast, to regulate anger, toddlers utilized an intermediate level of control (i.e., drawing the mother’s attention to themselves) and increased the use of this strategy with age. In addition, toddlers were able to select appropriate strategies for different stressors, and they increased with age the ability to adapt the strategies to the environmental conditions. Theoretical and practical implications are discussed.

## 1. Introduction

Emotion regulation (ER), defined as behaviors and strategies that serve to modulate the expression of emotion and affect, develops over the first years of life and is thought to be foundational for social adjustment in the preschool and school years [1]. According to Kopp’s developmental perspective, children have gone from responding only to highly arousing events and relying heavily on caregiver support as infants to being more active and purposeful as toddlers. Thus, they become more capable of overcoming impulsive reactions and, eventually, delay gratification. Finally, they reach a developmental endpoint [2] by manifesting a set of behaviors signaling the emergence of self-regulation at the beginning of preschool years. Two aspects, i.e., the ability to rely on their own rather than on others’ resources and to control their own reaction over prepotent responses [3], are involved in the passage from a more dependent to a more autonomous way of regulation [4,5,6]. Toddlerhood seems to play a key role in this transition; hence, every change in this short period from approximately 1 to 3 years of age represents a watershed in emotional development [7]. This study investigated emotion regulation at two time points in the toddlerhood period, i.e., 24 and 30 months, in anger and fear emotional eliciting contexts [8,9].

### 1.1. Emotion Regulation in Toddlerhood

Adaptive emotion regulation is a critical developmental goal underlying numerous psychological outcomes, including optimal cognitive performance in executive tasks, effective social strategies with peers, and management of stressful experiences in domestic and educational settings [10]. While this process begins at birth and continues through the entire life span, the first three years of life are of particular importance. During these years, children pass from exhibiting prepotent responses to highly arousing events to internally monitoring their emotional reactions and eventually manifesting a set of behaviors that involve the ability to self-regulate [2]. Toddlerhood is a stage where this transition can be observed, marking the passage from a more rigid to a more flexible way of regulation, with two abilities involved in this process.

The first one is the ability to rely on their own rather than others’ resources for regulation. Following Eisenberg and Morris [3], children reduced their reliance on extra organismic to use more intra-organismic regulation. During infancy, parents contribute as the dominant source of regulation. Through interactions with them in emotion-laden contexts, children learn over time that the use of strategies may be more useful for the reduction of emotional arousal than other strategies, thus becoming less dependent on others and more confident about their own regulatory skills [10,11,12]. Then, during toddlerhood, children show numerous self-oriented strategies to regulate negative affect. For example, toddlers may control their visual attention by engaging in frequent bouts of self-distraction or shifting focus to something else when faced with distressing stimuli [4,10,13]. They are also more likely than infants to direct their interactions with strangers [13] or to move away from fearful stimuli [2,13]. Toddlers also rely on their own skills in distressing contexts when searching for other people’s support. In this case, they elicit comfort from the adult through directed gaze or contact-seeking behaviors to obtain comfort during distressing situations; or they try to solve the situation through the adult by directing the adult’s attention or using information-seeking and social referencing behaviors [11,14]. Several studies suggested how the level of emotional distress determines whether the child will adopt a self-oriented or other-oriented strategy to regulate negative emotions, with toddlers appearing to preferentially ask for the caregiver’s help (i.e., other-oriented strategies) when highly distressed, as these behaviors appear to be effective (at least in part) in alleviating such distress [5,8,9]. Despite these studies referring to general distress without specifying which emotion was elicited, their results suggested toddlers tend to use specific strategies to cope with emotionally activating situations, showing greater flexibility according to the context and conditions of the stressor compared to behaviors characterizing regulating processes during infancy.

The second ability involved in ER development during toddler years is the ability to move from reactive to more purposeful responses. The advancements of motor and linguistic skills contribute to widening the set of regulatory strategies that are available for managing emotion in different situations [4,5,11], especially the advancement of cognitive and social cognitive skills, which is relevant for making children increasingly concerned with issues of control. Moreover, enhanced working memory, increased capacity for deductive reasoning, and greater ability for intentional communication contribute to children’s greater awareness of the contexts associated with their feelings. In this sense, children may have a greater ability to use more intentional and less automatic strategies to mitigate situational stresses and achieve specific goals [2,15,16,17]. Moreover, children change their emotional responses from stereotypical to flexible, from rigid to situationally responsive, and from over or under-arousing to performance-enhancing, thus achieving a greater ability to regulate their behavior to reduce, inhibit, amplify, and balance different affective responses [18]. The emergence of an ability to recruit internal resources for ER characterizes the developmental stage “self-control phase” [2]. In the model, this period covers the third year of life. It represents a major shift in development from the rigid responses (e.g., hesitation, wariness) that can be observed in the previous “control phase” to a more complex set of strategies (e.g., showing a delay in gratification, flexibility, and compliance with social norms) driven by an internally generated monitoring system, which characterizes the next “self-regulation phase”. Although previous studies have examined toddlers’ strategies in aversive contexts [8,9,16,19,20,21,22], there is a gap in the literature related to how these strategies evolve from other-oriented to self-oriented and from reactive response to more controlled strategies.

### 1.2. Emotion Regulation While Fear and Anger Situations

Emotion regulation strategies have been typically assessed in aversive contexts, as these situations require the greatest mobilization of regulatory resources to manage negative affect. According to the functionalist perspective [23], despite the negative emotions (e.g., fear or anger) having an adaptive function, they require regulation to achieve the goal for which the emotion was directed effectively. Fear is adaptive as its function is to avoid physical and psychological danger, thus propelling individuals to escape from threatening stimuli. Anger is adaptive, as it can motivate people to overcome the obstacles that hamper them from achieving their own goals. Toddlerhood is the developmental stage where specific emotion regulation strategies based on the emotional context become expressed. Leerkes and Wong [19] found that 16-month-old children were more likely to utilize self-soothing and “venting” strategies and less likely to use mother-oriented regulatory behaviors than when they experience other negative situations in fear-eliciting contexts. Moreover, the strategies more frequently used in fearful contexts, i.e., withdrawal [8], avoidance, and fussing to elicit the mother’s attention [9] revealed to serve an adaptive function by effectively mitigating children’s fear. Conversely, contradictory strategies, i.e., approaching and interacting with the stimulus [14] or playing with it, may not be adaptive in reducing fear because frightened children that tend to engage with the threatening stimulus do not achieve the goal of getting safe. When children are involved in anger-eliciting contexts, strategies of shifting attention away from the sources of frustration, passive waiting, and seeking information have been shown to attenuate the expression of anger. Conversely, strategies of focusing on the frustrating event were associated with an increase in the intensity of anger [15,24].

The literature highlighted that toddlers use more primitive strategies to manage fear while using more complex strategies to manage anger, comparing the aversive contexts eliciting fear and anger [19]. Diener & Mangelsdorf [9] found that avoiding stressful stimuli was associated with decreased expressions of fear but not anger, while venting behaviors (associated with the release of tension) had the opposite effect. Buss & Goldsmith [8] showed how withdrawal behaviors were likely to mitigate the expression of fear, whereas engagement behaviors with the object (e.g., struggling against the stimulus in the toy removal task) mitigated anger. In addition to the converging evidence suggesting a “qualitative” difference in effective regulation strategies between fear-inducing versus anger-inducing contexts, research also suggests a “quantitative” difference, with children using behavioral strategies more during the frustration context than during the fear context [9,21]. Although several studies have examined regulatory strategies toddlers use in contexts eliciting fear and anger, there is a gap regarding whether and how the two contexts elicit self-oriented versus other-oriented and reactive versus controlled regulatory strategies.

### 1.3. The Current Study

In line with the literature [3,25], the development of emotion regulation from infancy to toddlerhood is characterized by the children’s growing ability to select appropriate responses for different stressors, with relevant changes from other-regulated to autonomous strategies and from the use of reactive to more controlled strategies. To extend the literature, this study investigated the quality of toddlers’ emotion regulation strategies with respect to the two above aspects during two aversive contexts (i.e., fear and anger) at 24 and 30 months of age. Specifically, we coded emotion regulatory strategies as to whether they were self-oriented or other-oriented (i.e., asking whether children could regulate on their own or if they required the caregiver’s support) and whether they were reactive or controlled. In other words, if children reflected: (1) an automatic reaction to the negative event; (2) an attempt to manage the current emotion; or (3) an attempt to manage the context which can cause the emotion. We hypothesized that children would differ in their use of regulatory strategies based on the emotional context and age, expecting that toddlers will use self-oriented strategies more in fear than anger contexts and controlled strategies more at the older than younger age.

## 2. Method

### 2.1. Participants and Procedure

A total of 42 mother–child dyads participated in this study, recruited from the practices of family pediatricians in two urban areas in the middle of Italy. All children were Italian; 66% (*n* = 25) were male; 63% (*n* = 24) had one or more siblings, and 11% (*n* = 4) attended a daycare center. Children were considered for inclusion if they were healthy, with no developmental delays, according to their family pediatrician. Average maternal age was 35.17 years old (SD = 4.35; range = 26–45), and mothers had an average of 13.95 years of education (SD = 2.94; range = 8–18). All fathers and 76% of the mothers (*n* = 29) were employed with a stable job. Parents were informed that the aim of the study was to examine how children develop the ability to cope with emotional situations with or without their support, and their written consent was obtained. The families were not compensated for their participation, but they received a DVD with all videos collected at the lab. The sample was treated in accordance with the ethical standards outlined by the American Psychological Association and the Italian Association of Academic Psychologists, and the study was approved by the Department Ethics Review Board of G. D’Annunzio University of Chieti and Pescara. At 24 and 30 months of children’s age (+/−1 week of age), mothers were invited to the lab to participate in the Emotion Regulation Paradigm (cf. Section 2.2). Before the beginning of the procedure, the experimenter spent time in a small waiting room with the mother and child to become familiar with them. When mothers and their toddlers entered the laboratory room, the experimental procedures were explained. The room was outfitted with a table and two chairs (one for the child and one for the mother); only the experimental material was present.

### 2.2. Emotion Regulation Paradigm

Drawing from previous examples [9,21,26], a lab procedure composed of two contexts was implemented to elicit two basic emotions, respectively (fear and anger). Contexts were counterbalanced, with a positive moment of free play interposed between the two negative contexts (fear/anger). Each emotional context was 3 min in duration. The efficacy of the emotional stimuli in eliciting emotions had been previously tested in a pilot study with 24-month-old and 30-month-old children. Fear context. At 24 months of age, examiners used a remote-controlled toy snake with elements of novelty, unpredictability, and intrusiveness to examine children’s responses in a context that could elicit fear in the child. While the mother and the child were alone in the observation room, a remotely operated toy snake approached the child and moved forward rapidly to within 15 cm of the child and paused for 10 s. The snake then moved back and remained silent and stationary for another 10 s. The entire sequence was repeated one more time for a total of two trials. At 30 months of age, a spider with the same features of novelty and unpredictability was used. Both at 24 and 30 months, mothers were instructed to remain uninvolved and to refrain from assisting the child in managing their distress. During the procedure, the mother was seated on a couch and was provided with a magazine to read. If the child made bids for attention, the mother was allowed to respond with brief statements about the stimuli. The procedure was terminated if the child demonstrated high-intensity distress, which lasted greater than 30 s. Anger context. To examine children’s responses in a context where the child could elicit anger, the experimenter presented the children with an age-appropriate novel toy to play with, and after the child demonstrated engagement with the toy (playing with it for 1 min or more), the examiner took the toy away from the child, and placed it out of reach but within the child’s sight. At 24 months, the child was given a toy piano which had lights and sounds and contained colored balls inside; at 30 months, the child was given a toy candy dispenser. Mothers were instructed to remain uninvolved and to refrain from assisting the child in managing their distress. The paradigm was terminated if the child demonstrated high-intensity distress which lasted greater than 30 s.

### 2.3. Children’s Regulatory Behaviors’ Coding System

In line with previous studies which defined children’s behavioral strategies to regulate negative emotions in aversive contexts [8,19,27], we developed the following coding system to examine toddlers’ regulatory strategies. Strategies were grouped into two macro groups based on the ability of children to regulate on their own (self-oriented regulatory strategies) or to require the caregiver’s support (other-oriented regulatory strategies). Both groups of strategies were therefore divided into three levels indicating strategies reflecting: (1) an automatic reaction to the negative event (Self1 and Other1, respectively); (2) an attempt to manage the current emotion (Self2 and Other2, respectively); and (3) an attempt to manage the context which caused the emotion to occur (Self3 and Other3, respectively).

Self-Oriented Regulatory Strategies. Self 1: Tension release. The child shows repetitive vigorous motor behavior (waving arms, jumping, banging on the table), or instead, she/he freezes or retreats from the aversive stimulus (moves or turns away, tries to leave the room). Self 2: Self-soothing behaviors. The child actively engages in repetitive motor behaviors, such as sucking on fingers, twisting the hands or hair, scratching the head, and pulling the ear. Self 3: Dealing with the aversive situation. The child tries to engage with the stimulus, i.e., touching, handling, labeling, or speaking with the toy.

Other-Oriented Regulatory Strategies. Other 1: Fussing at the mother. The child demonstrates motor and/or emotional agitation while looking at the mother. Other 2: Drawing mother’s attention to themself. The child actively tries to engage the mother by looking at her and/or calling her. Other 3: Object/situational management by the mother. The child tries to engage the mother’s attention with the aim of solving the problem, e.g., pointing to the stimulus and looking at her alternatively, telling her something related to the actual event, and asking the mother to get the stimulus or for information about it.

The above regulatory strategies were coded dichotomously (present vs. absent) every 15 s in the anger and fear contexts from videotaped observations. All regulatory behaviors which occurred in the context were coded for each interval. The codes are exhaustive; that is, toddlers engaged in at least one of these behaviors during each 15 s interval. In addition, the codes are not mutually exclusive; that is, multiple codes could occur in each interval but not simultaneously. The scores for each of the regulatory behaviors were summed and divided by the total number of 15 s intervals in the context. The resulting proportion variables had a possible range of 0–1. Two separate trained coders coded all videos for fear and anger contexts, respectively. Interrater reliability was calculated based on ten cases at 24 months (25%) and nine cases at 30 months (23%) for each emotion regulation strategy in each context. The average Cohen’s Kappa value among strategies at 24 months was 0.94 (range: 0.92–0.96) and 0.95 (range: 0.89–0.96) in the fear context and anger context, respectively; at 30 months, it was 0.93 (range: 0.85–0.96) in the fear context and 0.92 (range: 0.82–96) in the anger context.

### 2.4. Analysis Plan

Preliminary analyses examined differences between males and females using the different emotion regulation strategies at each age and in each context using independent-sample t-tests. In addition, Pearson correlations were conducted to explore the associations between strategies at each age and across ages and contexts. The main analyses were run with the linear mixed model procedure of SPSS with REML as the method for estimation [28]. To represent variation that is due to individual differences, we entered intercepts and slopes for participants as random effects, with a variance components covariance structure. Visual inspection of residual plots did not reveal any obvious deviation from homoscedasticity or normality. The first set of analyses with linear mixed models explored the main effects of the Context (Anger vs. Fear), Age (24 months vs. 30 months), and their interaction on the relative frequency of each self-oriented emotion regulation strategy (Self 1, Self 2, Self 3). The second set of analyses with linear mixed models explored the main effects of the context (Anger vs. Fear), age (24 months vs. 30 months), and their interaction on the relative frequency of each other-oriented emotion regulation strategy (Other 1, Other 2, Other 3).

## 3. Results

### 3.1. Preliminary Results

The t-tests did not show any significant difference between males and females regarding the regulation strategy used in fear and anger contexts at both ages (24 and 30 months). This variable was not included in the other analysis. Correlations among the frequencies of self-oriented strategies are reported in Table 1. At 24 months, in the Fear context, a higher use of Self 3 was associated with a lower use of Self 1 and Self 2. In the Anger context, a higher Self 1 was associated with a lower Self 2 and a higher Self 2 with a lower Self 3. At 30 months, in the Fear context, a higher Self 2 was associated with a lower Self 1 and Self 3 and a higher Self 3 with a lower Self 1. Across ages, in the Anger context, a higher use of Self 2 at 24 months correlated with a higher use of the same strategy at 30 months. Across contexts, at 30 months, higher Self 2 in the Fear context correlated with higher Self 2 in the Anger context.

Correlations among the frequencies of other-oriented strategies are reported in Table 2. At 24 months, in the Fear context, a higher Other 2 was associated with a lower Other 3. At 30 months, in the Fear context, a higher Other 1 corresponded to a lower Other 2 and a higher Other 2 to a lower Other 3. In the Anger context, a higher Other 2 was associated with a lower Other 3. Across ages, in the Anger context, a higher Other 2 at 24 months corresponded with a higher use of the same strategy and a lower Other 3 at 30 months. Across contexts, at 24 months, a higher Other 3 in the Fear context was associated with a higher use of the same strategy in the Anger context. At 30 months, a higher Other 1 in the Fear context corresponded with a higher Other 3 in the Anger context, and Other 2 in the Fear context was associated with a higher Other 1 and a lower Other 3 in the Anger context. No significant associations were found between the other investigated variables.

### 3.2. Effects of Context and Age on Self-Oriented Emotion Regulation Strategies

The first set of mixed models examined the effects of context, age, and their interaction on self-oriented emotion regulation strategies. The two-way interaction between context and age was significant in the Self 1 strategy (cf. Table 3). Children demonstrated an increased use of Self 1 over age only in the Fear context, F = 167.39, *p* < 0.01, with a higher use of this strategy in the Fear than in the Anger context at 30 months, F = 18.18, *p* < 0.01. The main effects of age and context were significant in the Self 2 strategy, showing that the frequency of this strategy decreased over time in both contexts, and it was higher in the anger context than in the Fear context at both ages. Finally, the two-way interaction between context and age was significant in the Self 3 strategy. Children use this strategy more during Fear, F = 33.03, *p* < 0.01, than Anger, F = 9.23, *p* < 0.01, context with a decrease over age in both contexts, but stronger in the fear context (Fear context: F = 20.96, *p* < 0.01 and Anger context: F = 20.96, *p* < 0.01). Figure 1 shows the three interactions, respectively.

### 3.3. Effects of Context and Age on Other-Oriented Emotion Regulation Strategies

The second set of analyses explored the effects of context, age, and their interaction on other-oriented emotion regulation strategies (cf. Table 4). The two-way interaction between Context and Age was significant in the Other 1 strategy. The frequency of the Other 1 strategy increased over age only in the Fear context, F = 4.77, *p* = 0.03, with higher use in the Fear than in the Anger context at 30 months, F = 6.47, *p* = 0.02.

A significant main effect of context and Age on the Other 2 strategy was found. The Other 2 strategy was used more during the Anger than the Fear context, with a comparable increase over age. The two-way interaction between Context and Age was significant in the Other 3 strategy. The use of this strategy was higher in the Anger than in the Fear context at both ages (24 months: F = 5.01, *p* = 0.03; 30 months: F = 20.68, *p* < 0.01) with an increase of use in the Anger context over age, F = 7.98, *p* =< 0.01 (Figure 2).

## 4. Discussion

The literature suggested how the development of emotion regulation in early years is characterized by the children’s growing flexibility in regulating, signaled by an increasing ability to select appropriate strategies for different stressors [3]. In line with this, the current study examined the emotion regulation skills in two arousing contexts, such as fear and anger-inducing contexts, and during an age period, such as toddlerhood, where more flexible behaviors are beginning to emerge. We observed two categories of strategies as characterized by self-oriented and other-oriented behaviors, respectively, and we examined whether they represented active or controlled responses to the context, reflecting an automatic reaction to the current situation or an internal monitoring system. The aim was to examine changes in these strategies between 24 and 30 months in both the above aversive contexts. In line with the hypothesis, the results show how, independently of age, toddlers utilized self-oriented strategies to regulate emotions in the context of fear and other-oriented strategies to regulate emotions in the context of anger. This confirms how children are likely to rely on their own resources to manage fear [1,19] and on external assistance to manage anger [16]. The result also supports the functionalist perspective on the nature of emotion, which posits that emotions are goal-directed constructs [29] and, consequently, elicit strategies that meet the specific goal of the related emotional context. Since, in the context of fear, the primary goal is to maintain their own physical and psychological integrity, it is plausible that toddlers were likely to recruit their own resources to face a stimulus that is perceived as threatening by reacting immediately to it with the aim to escape from the situation and return to the homeostasis. On the other hand, in the context of anger, where the behavioral goal is to overcome the obstacles that interfere with their desired goal, toddlers were instead likely to engage in other-oriented strategies, eliciting the help of the caregiver to solve the situation, causing their frustration, i.e., to bring the desired object into proximity.

Partially in line with the hypothesis, the use of reactive vs. controlled emotion regulation strategies varied at 30 months compared to six months before, depending on the context. The results showed how toddlers were likely to use reactive self-oriented strategies (i.e., Self 1 strategies) at 30 months and more controlled strategies (Self 2 and Self 3 strategies) at earlier ages (i.e., at 24 months) in managing feelings in a fear context. While this may seem counter-intuitive, with older toddlers using more primitive strategies than younger toddlers, our findings align with previous research. Diener and Mangelsdorf [9] examined the behavioral strategies used to regulate the expression of fear in toddlerhood, showing that avoidance, coded when the child moves or turns away from the fearful stimulus, caused the feeling of fear to decrease more than by chance, thus proving to be an effective strategy to regulate emotions in a fearful context. In our coding schema, avoidance was one of the automatic reactions coded as Self 1 strategies and largely prevailed over the remaining others included in the same category (i.e., motor agitation, withdrawal, or freezing) for older toddlers when they were in the fearful context. In light of this, 30-month-old toddlers may have a greater awareness of the potential “dangerousness” of the fearful stimulus than six months before and, therefore, consistent with the functionalist approach. Their response was to avoid the threatening stimulus instead of interacting with it, as they were likely to do when younger. Since a reactive strategy is thought to be more effective for reducing fear than a controlled strategy, our finding suggests that toddlers regulate emotion in a more adaptive way when older than six months. Conversely, toddlers increased with age the use of controlled other-oriented strategies to regulate feelings of anger. Moreover, children increased Other 2 strategies (i.e., drawing the mother’s attention to help regulate the effect) more than Other 3 strategies (i.e., eliciting the caregiver’s attention to manage the frustrating context). We would expect that older toddlers were likely to improve the frequency of the most advanced other-oriented strategy (i.e., Other 3 strategy) by engaging the mother to obtain the desired object rather than the less advanced (i.e., Other 2 strategy), aimed to only obtain comfort from her. Methodological reasons could explain this finding. The experimental procedure prevented the children from accessing the attractive toy as well to the responses from the mother; it might be that the toddlers, recognizing that the mother was not available to remove the obstacle to the desired object, chose to have at least her comfort to mitigate the feelings of anger. It is also possible that we might have observed more advanced other-oriented strategies if the mother was allowed to be more responsive to the child’s bids and free to remove the source of frustration.

## 5. Conclusions

The study suggested that toddlers utilized self-oriented strategies to manage fear and utilized other-oriented strategies to manage anger, thus selecting appropriate strategies for different stressors. Moreover, we found that toddlers increased with age the strategies that were more effective in dealing with emotional demands and environmental conditions, although less advanced in the ability to monitor emotion internally. It suggests that the development of emotion regulation is signaled not simply by the cognitive level expressed by the regulatory strategies used at a given age but rather by their capacity to meet the goals of a given emotional situation. Both findings are in line with Buss [30], suggesting that the most important component of the immature pattern in emotion regulation is not a primitive response per se, but the lack of behavioral flexibility across contexts. In line with Brownell and Kopp [7], the flexibility of endpoints in demarcating toddlerhood recognize that development does not occur in a stepwise fashion from one month to the next and that there are no clear starting and ending points for any given competence, including emotion regulation.

Some limitations of the study need to be considered in the interpretation of the results and as directions for future research. First, due to the small sample size and the lack of a longitudinal assessment of ER strategies, including a third time point, such as 36 months of child’s age, we cannot generalize the results. Further studies should replicate our findings in a larger sample and a longer period of time. Second, we examined children’s strategies based on literature showing their role in emotion regulation. Therefore, due to our interest in the developmental change in these strategies across two different contexts, we did not assess how these strategies were effective in reducing the negative effect elicited by our paradigm. Future research should improve the investigation by testing the impact of these putative regulatory strategies on toddlers’ emotion regulation. Finally, despite a growing body of research indicating the importance of contextual effects on behavioral strategies (i.e., maternal involvement or children’s temperament), we measured the direct association between age and the strategy without examining potential mediating or moderator factors [31]. The lack of these data prevented the study from accounting for the response variability in a more complex way. Future studies could address this issue. The strength of this study was to examine self-regulation processes at two time points, i.e., at 24 and 30 months, in a crucial period for emotion regulation development corresponding to the self-control phase [2]. This study highlighted, indeed, the key role of age and context(s) in the advancing of flexible strategies during toddlerhood. Investigating these aspects is also relevant to define target training based on emotional regulation.

## Figures and Tables

**Figure 1 children-10-00878-f001:**
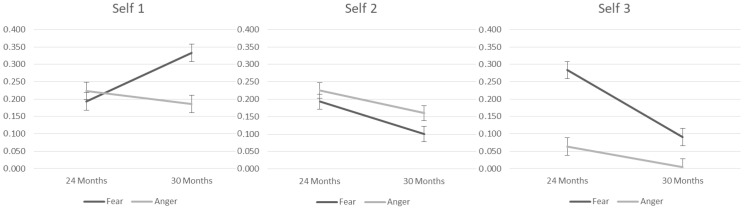
Mean predictive values of self-oriented emotion regulation strategies.

**Figure 2 children-10-00878-f002:**
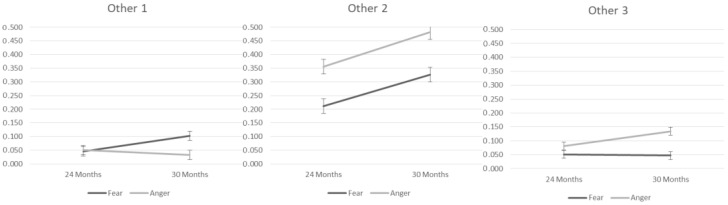
Mean predictive values of other-oriented emotion regulation strategies.

**Table 1 children-10-00878-t001:** Descriptives and correlations among frequencies of self-oriented strategies.

		M	SD	1	2	3	4	5	6	7	8	9	10	11	12
1	24 months Fear Self 1	0.19	0.15	-											
2	24 months Fear Self 2	0.19	0.13	0.21	-										
3	24 months Fear Self 3	0.28	0.26	−0.59 **	−0.37 *	-									
4	30 months Fear Self 1	0.33	0.19	0.15	−0.04	−0.26	-								
5	30 months Fear Self 2	0.10	0.14	−0.06	−0.04	0.11	−0.56 **	-							
6	30 months Fear Self 3	0.09	0.18	−0.14	0.01	0.28	−0.61 **	0.31 *	-						
7	24 months Anger Self 1	0.22	0.16	0.06	−0.20	0.30	0.15	−0.01	−0.03	-					
8	24 months Anger Self 2	0.23	0.16	−0.06	0.12	−0.13	−0.16	0.30	0.25	−0.46 **	-				
9	24 months Anger Self 3	0.06	0.07	−0.15	0.04	0.28	−0.14	−0.25	0.19	0.18	−0.44 **	-			
10	30 months Anger Self 1	0.19	0.11	0.03	0.16	0.07	−0.09	0.07	0.23	0.25	−0.01	0.16	-		
11	30 months Anger Self 2	0.16	0.13	0.10	0.11	−0.17	−0.08	0.41 **	0.07	−0.34 *	0.62 **	−0.27	−0.08	-	
12	30 months Anger Self 3	0.01	0.01	−0.11	0.10	−0.17	−0.22	0.23	−0.08	−0.22	0.25	−0.12	0.03	0.25	-

Note: * *p* < 0.05, ** *p* < 0. 01.

**Table 2 children-10-00878-t002:** Descriptives and Correlations among Frequencies of Other-Oriented Strategies.

		M	SD	1	2	3	4	5	6	7	8	9	10	11	12
1	24 months Fear Other 1	0.05	0.11	-											
2	24 months Fear Other 2	0.21	0.17	−0.05	-										
3	24 months Fear Other 3	0.05	0.08	−0.01	−0.31 *	-									
4	30 months Fear Other 1	0.10	0.15	0.16	−0.01	0.08	-								
5	30 months Fear Other 2	0.33	0.21	−0.01	0.01	0.16	−0.44 **	-							
6	30 months Fear Other 3	0.05	0.07	−0.14	−0.19	0.00	−0.04	−0.37 *	-						
7	24 months Anger Other 1	0.05	0.06	−0.19	0.03	0.00	−0.08	0.21	0.06	-					
8	24 months Anger Other 2	0.36	0.14	0.26	0.16	−0.05	0.15	0.07	−0.12	−0.22					
9	24 months Anger Other 3	0.08	0.07	−0.03	−0.13	0.31 *	0.19	0.16	0.04	0.14	0.06	-			
10	30 months Anger Other 1	0.03	0.10	−0.09	−0.17	−0.09	0.02	0.33 *	0.01	0.18	−0.13	0.10	-		
11	30 months Anger Other 2	0.48	0.16	0.19	0.26	−0.23	−0.06	0.23	−0.24	−0.05	0.50 **	0.12	−0.08	-	
12	30 months Anger Other 3	0.13	0.12	−0.14	−0.16	0.16	0.44 **	−0.41 **	0.26	0.22	−0.45 **	0.30	−0.16	−0.42 **	-

Note: * *p* < 0.05, ** *p* < 0. 01.

**Table 3 children-10-00878-t003:** Mixed model results in self-oriented emotion regulation strategies.

	Self 1		Self 2		Self 3	
	F	*p*	F	*P*	F	*p*
Intercept	287.97	<0.01	137.77	<0.01	55.20	<0.01
Context	6.16	0.01	6.09	0.02	42.40	<0.01
Age	4.89	0.03	18.03	<0.01	28.67	<0.01
Context × Age	14.59	<0.01	0.53	0.47	7.97	0.01

**Table 4 children-10-00878-t004:** Mixed model results on other-oriented emotion regulation strategies.

	Other 1		Other 2		Other 3	
	F	*p*	F	*P*	F	*p*
Intercept	47.08	<0.01	426.09	<0.01	87.97	<0.01
Context	3.67	0.06	38.65	<0.01	22.85	<0.01
Age	1.39	0.24	25.35	<0.01	4.01	0.05
Context × Age	5.11	0.03	0.06	0.81	5.42	0.02

## Data Availability

Data can be requested by the corresponding author.

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
