# Peer review of "Emotion Regulation in Toddlerhood: Regulatory Strategies in Anger and Fear Eliciting Contexts at 24 and 30 Months"

_children, 2023, doi:10.3390/children10050878_

Round 1

Reviewer 1 Report

The issue is important and the article is well constructed, the design and analysis were well performed, and the result are clearly presented. 

I have only few minor comments:

-          Page 5 line 191 – ‘implemented to elicit three basic emotions respectively (fear and anger)’- please rephrase and name the three emotions mentioned.

-          Page 6 line 280 – ‘higher use of Self 3 was associated with a lower use of Self 1 and Self 3” - it is my understanding that instead Self 3 it should be Self 2

         Page 9 line 355, 356 – “self-oriented strategies to regulate feelings” - I would suggest replacing the term feelings with the term emotions, the term feelings implying usually a more elaborate type of emotions or affect.

-          Page 11 line 421 – “the results are we ca not generalized the results” -please correct the sentence.

-          Page 11 line 422 – “replicate our findings in a larger and longer sample” – I believe that larger sample and longer period of time would be more appropriate.

-          Children exhibit differences in behavioural strategies across the emotion eliciting situations as a function of maternal involvement. The growing body of research indicate the importance of contextual effects on behavioural strategies.

Author Response

Dear Reviewer,

Thank you for your kind feedback and for the interesting comments to our manuscript. We modified the paper following your suggestions.

The issue is important and the article is well constructed, the design and analysis were well performed, and the result are clearly presented. 

I have only few minor comments:

Page 5 line 191 – ‘implemented to elicit three basic emotions respectively (fear and anger)’- please rephrase and name the three emotions mentioned.

We are sorry, that was a typing error, the correct sentence is the following: Drawing from previous examples [9, 21,26], a lab procedure composed by two contexts was implemented to elicit two basic emotions respectively (fear and anger).

Page 6 line 280 – ‘higher use of Self 3 was associated with a lower use of Self 1 and Self 3” - it is my understanding that instead Self 3 it should be Self 2

We are sorry,we corrected the error, we meant Self 2. We corrected the sentence.

Page 9 line 355, 356 – “self-oriented strategies to regulate feelings” - I would suggest replacing the term feelings with the term emotions, the term feelings implying usually a more elaborate type of emotions or affect.

Thank you for the suggestion, we replaced the term as suggested:

In line with the hypothesis, the results show how, independently of the age, toddlers utilized self-oriented strategies to regulate emotions in the context of fear, and other–oriented strategies to regulate emotions in the context of anger.”

Page 11 line 421 – “the results are we ca not generalized the results” -please correct the sentence.

The correct sentence is the following:

“Due to the small sample size and the lack of a longitudinal assessment of ER strategies including a third time point, such as 36 months of child’s age, we can’t generalize the results.”

Page 11 line 422 – “replicate our findings in a larger and longer sample” – I believe that larger sample and longer period of time would be more appropriate.

We corrected the sentence as suggested:

“Further studies should replicate our findings in a larger and longer period of time.”

Children exhibit differences in behavioural strategies across the emotion eliciting situations as a function of maternal involvement. The growing body of research indicate the importance of contextual effects on behavioural strategies.

This point is now better specified in the conclusions section as follows:

“Finally, despite a growing body of research indicating the importance of contextual effects on behavioral strategies (i.e. maternal involvement or children’s temperament), we measured the direct association between age and the strategy without examining potential mediating or moderator factors [3]. The lack of these data prevented the study to account for the response variability in a more complex way. Future studies could address this issue.”

Reviewer 2 Report

I found this work interesting, well-developed, and significant for the field.

I suggest a few points that may be further improved.

In lines 40-43, the aim of the study is announced while the authors are still introducing the background data of the literature. Thus, I think that it is unnecessary to anticipate the aim of the work, which is extensively drawn in lines 148-164, where all the background data have been given to the readers.

What is stated in lines 63-64 is quite unclear: which strategies are better or worse than others? I think it should be better explained.

Line 191: maybe “two” instead of “three” basic emotions

Along the manuscript, a few minor mistakes should be corrected (lines 44-45, 421, and 432)

Author Response

Dear Reviewer thank you very much for your appreciations to our work. We modified the paper as suggested.

I found this work interesting, well-developed, and significant for the field.

I suggest a few points that may be further improved.

In lines 40-43, the aim of the study is announced while the authors are still introducing the background data of the literature. Thus, I think that it is unnecessary to anticipate the aim of the work, which is extensively drawn in lines 148-164, where all the background data have been given to the readers.

Thanks for the suggestion, we modified the sentence as follows:

“The present study investigated the emotion regulation at two time points in the toddlerhood’s period, i.e., 24 and 30 months in anger and fear emotional eliciting contexts [8,9].”

What is stated in lines 63-64 is quite unclear: which strategies are better or worse than others? I think it should be better explained.

We agree with the reviewer, the sentence was modified to clarify the meaning:

“Despite these studies referred to general distress without specifying which emotion was elicited, their results suggested toddlers tend to use specific strategies to cope with the emotional activating situations, showing greater flexibility according to the context and conditions of the stressor compared to behaviors characterizing regulating processes during infancy.”

Line 191: maybe “two” instead of “three” basic emotions

We are sorry, that was a typing error, the correct sentence is the following: Drawing from previous examples [9, 21,26], a lab procedure composed by two contexts was implemented to elicit two basic emotions respectively (fear and anger).

Along the manuscript, a few minor mistakes should be corrected (lines 44-45, 421, and 432)

Thank you, we corrected the mistakes along the manuscript.